# Evolving Role of Immunotherapy in Advanced Biliary Tract Cancers

**DOI:** 10.3390/cancers14071748

**Published:** 2022-03-29

**Authors:** Sandra Kang, Bassel F. El-Rayes, Mehmet Akce

**Affiliations:** 1Department of Hematology and Medical Oncology, Winship Cancer Institute, Emory University, Atlanta, GA 30322, USA; smkang3@emory.edu; 2Department of Internal Medicine, Division of Hematology and Oncology, O’Neal Comprehensive Cancer Center, University of Alabama at Birmingham Heersink School of Medicine, Birmingham, AL 35233, USA; belrayes@uabmc.edu

**Keywords:** biliary tract cancer, cholangiocarcinoma, gallbladder cancer, immunotherapy, immune checkpoint inhibitors, tumor microenvironment

## Abstract

**Simple Summary:**

Biliary tract cancers (BTC) include gallbladder cancers, intrahepatic, perihilar and distal extrahepatic cholangiocarcinomas. BTCs represent a major health problem due to their increasing global incidence and associated poor prognosis. The majority of patients present with advanced stages of cancer, where cytotoxic chemotherapy provides modest survival benefit. More recently, novel treatment options have emerged with the development of agents targeting specific genetic mutations of tumors as well as immunotherapy, which enhances the immune system’s ability to target cancer cells efficiently. In this review, we will discuss current and emerging systemic therapy options and the rationale for immunotherapy in BTC.

**Abstract:**

Biliary tract cancers (BTC) comprise a rare and diverse group of malignancies that involve the gallbladder and biliary tree. These cancers typically present in later stages because they are aggressive in nature and affected patients are often asymptomatic in earlier stages of disease. Moreover, BTCs are generally refractory to cytotoxic chemotherapy, which further contributes to their associated poor survival outcomes. Novel therapy approaches are clearly needed. Molecular targeted agents have been developed based on our expanding knowledge of the genetic mutations underlying BTCs and represent a promising treatment strategy in molecularly selected subgroups of patients. In addition, the advent of immunotherapy over recent years has dramatically changed the bleak outcomes observed in malignancies such as melanoma. Our growing understanding of the complex tumor microenvironment in BTC has identified mechanisms of tumor immune evasion that could potentially be targeted with immunotherapy. As a result, different immunotherapeutic approaches including immune checkpoint inhibitors, cancer vaccines, and adoptive cell therapy, have been investigated. The use of immunotherapeutic agents is currently only approved for a small subset of treatment-refractory BTCs based on microsatellite instability (MSI) status and tumor mutational burden (TMB), but this will likely change with the potential approval of immunotherapy plus chemotherapy as a result of the TOPAZ-1 trial.

## 1. Introduction

Biliary tract cancers (BTC) are a heterogeneous group of rare but highly lethal carcinomas that originate from the gallbladder, intrahepatic bile ducts, and extrahepatic (perihilar and distal) bile ducts [1]. Cholangiocarcinoma (CCA), or cancer arising from the bile ducts, accounts for 3% of all gastrointestinal malignancies and is the second most common primary liver cancer after hepatocellular carcinoma (HCC). Incidence of CCA varies greatly by geographic region, with the highest rates observed in Southeast Asia, including Thailand, where cases are as high as 113 per 100,000 in men and 50 per 100,000 women [2]. Although incidence rates are about 40-fold lower in Western regions, the prevalence of intrahepatic CCA (iCCA) has been increasing in most countries in recent decades. The majority of CCA cases are sporadic, but there are a number of established risk factors including chronic parasitic infection with liver flukes, primary sclerosing cholangitis, hepatolithiasis, choledochal cysts, and chronic liver disease caused by cirrhosis and viral hepatitis [2,3]. Gallbladder carcinoma (GBC), on the other hand, is most prominent in South American countries and is strongly linked to cholelithiasis [1]. Despite differences in the pathophysiology of GBC and CCA, they are often grouped together and management for both, especially in advanced stages, is similar. 

Surgery represents the only potentially curative treatment option for earlier stages of BTC. Unfortunately, most patients (70%) initially present with unresectable or metastatic disease, which is associated with dismal 5-year survival rates of less than 5% [4,5]. Select patients with unresectable but nonmetastatic disease may be considered for locoregional therapies including transarterial chemoembolization (TACE), hepatic arterial infusion (HAI) of chemotherapy, radioembolization with yttrium-90 (Y-90), or radiotherapy. However, the use of locoregional therapy in advanced BTC has not yet been established in prospective randomized control trials [6]. Systemic therapy is, therefore, the mainstay of treatment for advanced BTC. The landmark phase III ABC-02 trial established the combination of gemcitabine and cisplatin (GEMCIS) as the standard first-line treatment in advanced settings, although median overall survival (OS) remains poor at less than 1 year [7]. Other chemotherapy regimens in first- and later-line settings have also shown only modest survival benefits, emphasizing the unmet need to improve therapeutic options. In recent years, increased understanding of the molecular biology of BTC has led to the development of novel targeted agents against tumors harboring certain molecular alterations, including fibroblast growth factor receptor-2 (FGFR2) fusions and isocitrate dehydrogenase-1 (IDH1) mutations [8]. Promising results from prospective clinical trials have led to FDA approval of targeted agents such as pemigatinib and ivosidenib for FGFR2 fusion or other rearrangement-positive and IDH1-mutated CCA, respectively, after progression on at least one line of therapy [9,10,11]. Table 1 summarizes the key findings of clinical trials of currently approved systemic therapies for advanced BTC.

Immunotherapy has significantly expanded the scope of cancer treatment in recent years. The success of immune-directed approaches in malignancies such as melanoma and renal cell carcinoma has sparked interest in evaluating their use in BTC [15,16]. Since the pathogenesis of BTC is often associated with chronic inflammation, studies have examined the BTC tumor microenvironment (TME) to identify potential targets of immunotherapy. Growing evidence has shown that immune checkpoint inhibitors (ICIs) are effective and these are FDA-approved in a small subset of BTC patients at present, including those with DNA mismatch repair-deficient (dMMR) or microsatellite-instability-high (MSI-H) and tumor-mutational-burden-high (TMB ≥ 10 mutations/megabase) tumors in treatment-refractory settings [17,18]. However, based on recent announcement of improved survival with durvalumab plus GEMCIS compared to GEMCIS alone in the TOPAZ-1 trial, ICI-based systemic therapy is poised to become a new frontline therapy option, regardless of TMB and MMR/MSI status [19]. Adoptive cell therapy and cancer vaccines are other immunotherapeutic strategies that are being evaluated in BTC [20]. In this review, we will discuss current and emerging systemic therapy options and review the role of ICIs as well as other BTC immunotherapy approaches that are being investigated in ongoing studies.

## 2. Current Treatment Paradigm of Advanced BTC

### 2.1. First-Line Systemic Therapy

GEMCIS is the current standard of care first-line treatment for advanced BTC. In the phase III ABC-02 trial, 410 patients with locally advanced or metastatic BTC were randomized to GEMCIS or gemcitabine alone. Doublet chemotherapy had superior survival outcomes compared to monotherapy, with a median OS of 11.7 months versus 8.1 months (hazard ratio [HR] 0.64, *p* < 0.001), median progression-free survival (PFS) of 8 months versus 5 months (*p* < 0.001), and a disease control rate (DCR) of 81.4% versus 71.8% (*p* = 0.049). Toxicity profiles were similar between both groups except for neutropenia, which was more frequent in the combination treatment arm, although not associated with higher rates of neutropenic fever or infection [7]. The ABC-02 trial findings were supported by the Japanese randomized phase II BT22 study that also demonstrated an improved median OS of 11.2 months with GEMCIS compared to 7.7 months with gemcitabine monotherapy [21]. 

Given the modest survival benefit with GEMCIS, many trials have attempted to develop new first-line treatment strategies. The phase III FUGA-BT trial showed noninferior survival outcomes with gemcitabine plus oral fluoropyrimidine S-1 (tegafur, gimeracil, oteracil) compared to GEMCIS in 354 Japanese patients with chemotherapy-naïve recurrent or unresectable BTC (median OS 15.1 months with S-1 vs. 13.4 months with cisplatin, HR 0.95). Both treatments were generally well tolerated, but adverse events, including cytopenias and peripheral neuropathy, were more common with cisplatin, whereas oral mucositis and diarrhea were more frequent with S-1 [12]. Although S-1 is available in many Asian countries, its application in Western countries, including the United States, is not as widespread due to the increased gastrointestinal toxicity observed in Caucasian compared to East Asian patients [22]. Gemcitabine plus oxaliplatin (GEMOX) could be another reasonable alternative for first-line therapy in cisplatin-ineligible patients based on multiple phase II studies [23,24,25]. In one phase III trial, median OS with GEMOX versus GEMCIS was numerically higher with the former regimen (9 months vs. 8.3 months, *p* = 0.057), but did not meet the criteria for equivalence [26]. Capecitabine plus oxaliplatin (CAPOX) was noninferior to GEMOX for 6-month PFS rates in a randomized phase III study, but has not been directly compared to first-line GEMCIS [27]. Other doublet regimens that have shown efficacy and safety in phase II studies include gemcitabine plus capecitabine [28], gemcitabine plus nab-paclitaxel [29], and nanoliposomal-irinotecan (nal-IRI) plus 5-fluorouracil (5-FU)/leucovorin [30]. The phase Ib ABC-08 study showed encouraging efficacy using the combination of cisplatin and novel agent NUC-1031, a phosphoramidate transformation of gemcitabine designed to overcome drug resistance [31], prompting the ongoing phase III NuTide:121 trial comparing this regimen to GEMCIS (NCT04163900). 

More intensive triplet therapy regimens have also been investigated in the first-line setting for advanced BTC. Modified FOLFIRINOX (5-FU, irinotecan, oxaliplatin) failed to improve 6-month PFS compared to GEMCIS (44.6% vs. 47.3%) in the phase II/III PRODIGE 38 AMEBICA trial [32]. In Japan, GEMCIS plus S-1 improved survival outcomes compared to GEMCIS in 246 patients with advanced BTC, with a median OS of 13.5 months versus 12.6 months (HR 0.79, *p* = 0.046) [33]. Promising results from a phase II study [34] of the combination of gemcitabine, nab-paclitaxel, and cisplatin (GAP) that demonstrated a remarkable median OS of 19.2 months, although with frequent rates (58%) of grade 3 or higher toxicity, provide a basis for the ongoing phase III SWOG-1815 trial comparing this triplet regimen with GEMCIS (NCT03768414). In general, alternative cytotoxic chemotherapy regimens have not significantly improved on GEMCIS, which remains the preferred first-line systemic therapy. This is likely to change with the potential approval of durvalumab plus GEMCIS regimen, as discussed in Section 4.2.2 [19].

### 2.2. Systemic Treatment beyond First-Line Therapy

#### 2.2.1. Cytotoxic Chemotherapy-Based Approach

Due to the common decline in performance status after progression with first-line therapy, only 15–25% of patients are able to receive second-line chemotherapy [8]. Data on chemotherapy options after first-line treatment are limited. A systematic review of 25 studies, including phase II trials, retrospective studies, and case reports, showed a median OS of 7.2 months with second-line treatment compared to an estimated 4 months with the best supportive care only. The authors cautioned interpretation of these results given the selection bias of fitter patients eligible for second-line chemotherapy, and additional subgroup analysis of the included phase II trials showed inferior survival outcomes compared to those in retrospective studies (median OS 6.6 vs. 7.7 months) [35]. More recently, the phase III ABC-06 trial randomized 162 patients with BTC, who progressed on first-line GEMCIS to active symptom control (ASC) with or without FOLFOX (5-FU/leucovorin and oxaliplatin). Although differences in the primary endpoint of median OS were modest (6.2 vs. 5.3 months; HR 0.69, *p* = 0.031), FOLFOX resulted in clinically meaningful improvements in survival rates at 6 months (50.6% vs. 35.5%) and 12 months (25.9% vs. 11.4%), establishing this regimen as a second-line chemotherapy option. However, whether FOLFOX is superior to fluoropyrimidine alone has not been verified and is an important question to address given the high rates of grade 3 or 4 toxicity (59% with FOLFOX plus ASC vs. 37% with ASC alone) [13].

#### 2.2.2. Molecularly Selected Targeted Therapy Approaches

The growing availability of molecular profiling has revealed significant genetic heterogeneity among BTCs that vary depending on the anatomic location of the primary tumor. Some molecular aberrations have been identified as potential therapeutic targets, leading to the development of novel targeted agents that are currently used beyond the first-line setting. IDH1 and IDH2 are key enzymes involved in glucose metabolism. The increased activity of these enzymes results in accumulation of the oncometabolite 2-hydroxyglutarate (2-HG), resulting in impaired DNA repair and the promotion of carcinogenesis [36]. Mutations of IDH1 and IDH2 are detected in 10–20% and 3% of BTCs, respectively, predominantly in iCCA [37]. Ivosidenib, an oral small-molecule inhibitor of IDH1, demonstrated effectiveness and safety for IDH1-mutated advanced BTC in the phase III ClarIDHy trial. A total of 185 patients who had progressed to two lines of therapy were randomized to ivosidenib (500 mg once daily in continuous 28-day cycles) or placebo. The primary endpoint median PFS was significantly improved with ivosidenib compared to placebo (2.7 vs. 1.4 months, HR 0.37, one-sided *p* < 0.0001) and updated analysis showed a trend towards better median OS in the experimental arm (10.3 vs. 7.5 months, HR 0.79, *p* = 0.09), which may have been impacted by crossover from the placebo to treatment group [10,38]. Overall responses with ivosidenib were low at 2% (3/124, all partial responses [PR]), but more than half of patients (51%) in this treatment arm had stable disease, while no patients on the placebo achieved a response and had a DCR of only 28%. The most common adverse events related to ivosidenib were fatigue, nausea, and diarrhea [10]. These results led to the FDA approval of ivosidenib for previously treated, locally advanced or metastatic IDH1-mutated BTC [39]. In addition, ivosidenib, studies evaluating other agents targeting IDH1/2-mutated BTCs are ongoing (NCT02073994, NCT04521686, NCT02746081). 

FGFR2 fusions and rearrangements result in constitutively active tyrosine kinases, which activate downstream signaling pathways, including MAPK, that promote cell proliferation and survival. Up to 20% of iCCAs harbor FGFR2 translocations [1]. In the phase II FIGHT-202 study, 146 patients with previously treated, advanced cholangiocarcinoma with and without FGFR2 alterations received pemigatinib (13.5 mg once daily 2 weeks on, 1 week off in 21-day cycles), an oral inhibitor of FGFR1-3. Those with FGFR2 alterations achieved an objective response rate (ORR) of 38% with three complete responses (CR) and a DCR of 80%, whereas no patients with other or without FGFR mutations achieved a response. Pemigatinib was overall well-tolerated, and the most common all-grade side effect was hyperphosphatemia [9]. Other FGFR inhibitors, including infigratinib [14], futibatinib [40], derazantinib [41], Debio 1347 [42], and erdafitinib [43], have also demonstrated promising anti-tumor activity in phase I/II trials. Positive results led to FDA approval of pemigatinib and infigratinib and breakthrough therapy designation of futibatinib for previously treated FGFR2-altered BTC. Phase III trials comparing these FGFR inhibitors against GEMCIS as first-line therapy in advanced BTC are underway (pemigatinib in FIGHT-302 [NCT03656536]; infigratinib in PROOF [NCT03773302]; futibatinib in FOENIX-CCA3 [NCT04093362]). 

Epidermal growth factor receptor (EGFR) mutations have been described in 15% of BTC patients and may trigger signaling cascades that contribute to cancer development and progression [44]. Unfortunately, targeted agents against EGFR, including erlotinib, cetuximab, and panitumumab, have overall failed to show significant survival benefit both as a monotherapy and in combination with chemotherapy in advanced BTC [45,46,47]. Human epidermal growth factor receptor 2 (HER2) amplification is seen in up to 19% of BTC, primarily in GBC (16%), followed by extrahepatic cholangiocarcinoma (eCCA) (11%) and iCCA (3%) [10]. HER2-directed therapy with trastuzumab plus pertuzumab and neratinib has shown clinical activity for relapsed and advanced BTC in previous phase II studies [48,49]. MyPathway was a non-randomized phase II basket study that enrolled 39 patients with previously treated metastatic BTC with HER2 amplification and/or overexpression to receive pertuzumab (840 mg loading dose, then 420 mg every 3 weeks) plus trastuzumab (8 mg/kg loading dose, then 6 mg/kg every 3 weeks). The ORR was 23% (9/39, all PR), with a median duration of response of 10.8 months and no treatment-related deaths [48]. Further trials will be needed to confirm the survival benefit of HER2-directed treatment in this patient population. 

BRAF V600E mutations are detected in approximately 5% of iCCA and serve as another potential therapeutic target in advanced BTC [7]. Forty-three patients with previously treated advanced BTC harboring BRAF V600E mutations were treated with the combination of BRAF inhibitor dabrafenib (150 mg twice daily) plus MEK inhibitor trametinib (2 mg once daily) in the phase II ROAR trial. At a median follow-up of 10 months, overall response was achieved in 47% of patients with a median PFS of 9 months. The most common treatment-related side effects were fever, rash, nausea, and diarrhea [50]. 

Neurotropic tyrosine receptor kinase (NTRK) rearrangements are rare in BTC, with a reported incidence of 3.5% in iCCA. TRK inhibitors larotrectinib and entrectinib resulted in a high ORR (57–75%) in NTRK fusion-positive advanced solid tumors including CCA [51,52]. The phase II STARTRK-2 basket study (NCT02568267) of entrectinib for CCA and other solid tumors harboring NTRK, as well as ROS1 and anaplastic lymphoma kinase (ALK) rearrangements, is ongoing. ALK and ROS1 gene fusions are uncommon in BTC, with reported rates of 2.6% and 8.7%, respectively [53,54]. To date, the data on targeted agents such crizotinib targeting ALK and ROS1 fusions in BTC are extremely limited and additional investigation is required [55,56].

The PI3K/AKT/mTOR pathway is a key regulator of cell metabolism, growth, and survival. Mutations causing dysregulation of this pathway have been implicated in BTC and provide a basis for targeted therapies in this setting. The mTOR inhibitor everolimus (10 mg once daily) was evaluated as a first- and second-line therapy for advanced BTC in the phase II RADiChol [57] and ITMO [58] trials, respectively. These studies demonstrated a similar DCR of 45–48% and ORR of 5–12% [57,58]. Previously untreated patients achieved a median PFS of 5.5 months and median OS of 9.5 months with everolimus as a first-line treatment. Biomarker analysis of the PI3K/AKT/mTOR pathway using immunohistochemical staining did not correlate with clinical outcomes [57]. Results from early studies of PI3K inhibitors (buparlisib [59] and copanlisib [60]) and AKT inhibitors (MK2206 [61]) in BTC have been modest and data are limited.

In summary, targeted therapies overall represent a major advancement in treatment for BTC and additional treatment options will likely grow with the identification of novel driver mutations. As more patients receive these targeted agents, many will inevitably develop acquired drug resistance and, consequently, new therapeutic strategies to overcome resistance will need to be explored. 

## 3. Tumor Microenvironment of BTC

BTCs are generally characterized by a desmoplastic tumor microenvironment (TME), composed primarily of stromal cancer-associated fibroblasts (CAFs) as well as immune cells including tumor-associated macrophages (TAMs), natural killer (NK) cells, and T cells [62]. CAFs are activated myofibroblasts of unclear origin, but potentially derive from hepatic stellate cells, portal fibroblasts, and bone marrow-derived mesenchymal cells. When activated by cancer cells via factors such as platelet-derived growth factor (PDGF), CAFs, in turn, extensively remodel the extracellular matrix (ECM) that renders tumor tissue stiffer and consequently triggers signaling pathways, which ultimately contribute to cancer development and progression. CAFs also secrete several cytokines, chemokines, and growth factors, including fibroblast growth factor (FGF) and transforming growth factor (TGF)-β, which promote tumor invasion and survival [63]. Elevated expression of the CAF phenotypic marker α-smooth muscle actin has been associated with poor survival outcomes in patients with CCA [64]. Furthermore, preclinical studies demonstrated that the depletion of CAFs results in decreased tumor growth, lymphatic vascularization, and metastasis in CCA mouse models, highlighting the importance of CAFs in carcinogenesis [65]. 

Activated CAFs promote an overall immunosuppressive TME in BTC that permits tumor immune evasion and progression by regulating both innate and adaptive immunity. Regarding innate immunity, CAFs recruit immune cells, including TAMs and myeloid-derived suppressor cells (MDSCs), and reduce the activation of NK cells. Cancer cells stimulate the polarization of TAMs towards the pro-tumor M2 phenotype, which leads to the production of factors such as vascular endothelial growth factor (VEGF)-A and cyclooxygenase (COX)-2, which promote angiogenesis and tumor growth [63,66]. MDSCs are immature myeloid cells that induce immunosuppressive regulatory T cell (Treg) expansion and inhibit cytotoxic NK cells by the secretion of cytokines such as interleukin (IL)-10 and TGF-β [67]. Although the role of NK cells in BTC is not well-defined, a study demonstrated that the infusion of ex vivo-expanded human NK cells in xenograft mouse models resulted in significant CCA growth inhibition, suggesting an important role of NK cells in the anti-tumor immune response [68]. 

Dendritic cells (DCs) are critical for the induction of adaptive immunity through antigen presentation to activate effector cells. CAFs attract DCs and diminish their expression of antigen-presenting HLA molecules, resulting in reduced activation of tumor-infiltrating lymphocytes (TILs) [63,69]. TILs include CD8+ cytotoxic T lymphocytes and CD4+ T helper lymphocytes that function to identify and target cancer cells, as opposed to Tregs, which suppress the immune system to maintain self-tolerance [68]. Multiple studies have confirmed that increased CD4+ and CD8+ T cell infiltrates are associated with less aggressive disease and improved survival outcomes, whereas lower levels of these cells and higher expression of Tregs correlate with poor survival in CCA patients [70,71,72]. Tregs are distinguished by the overexpression of forkhead box P3 (FoxP3), a transcription factor often accompanied by the upregulation of cytotoxic T lymphocyte antigen 4 (CTLA-4) [73]. CTLA-4 and other immune checkpoints, such as programmed death-1 (PD-1), are frequently exploited through overactivation by their specific ligands (e.g., programmed death ligand 1 [PD-L1]) that are expressed on cancer and immune cells, leading to peripheral T cell exhaustion, and thus allowing for tumor escape from immune surveillance [62]. 

With our growing understanding of the underlying BTC TME, significant heterogeneity in TME has been found among these patients. One study examined 78 iCCA tumors and identified four immune subtypes (immune desert, immunogenic, myeloid, and mesenchymal) in the TME. These TME subtypes differed in their composition and abundance of infiltrating cells such as CAFs, myeloid cells, and effector T cells, as well as the gene expression of immune signaling pathways. The majority (45%) of tumors displayed an immune desert phenotype characterized by the depletion of HLA molecules and reduced immune activation in contrast to those in the immunogenic subgroup, which demonstrated an inflammatory TME enriched with immune cells. As a result, the immune classification of TME in BTCs may correspond to distinct immune escape mechanisms and, therefore, their different responses to immunotherapy, which could help identify the patients that would benefit most from immunotherapeutic approaches [74]. Another potential strategy used in many other cancer types is the use of predictive biomarkers. Previous studies have reported PD-L1 expression in 9–72% of BTC patients, with higher levels correlated to more invasive disease and poor survival but better responses to immunotherapy in select cases [75,76,77,78,79,80]. High TMB has been reported in 5.9% of BTC patients, among whom 36% also demonstrated microsatellite instability (MSI)/deficient mismatch repair (dMMR), in a whole-exome sequencing study of 231 CCA tumor samples [81]. However, these markers often do not correlate with clinical outcomes in BTC patients, as later discussed in Section 4, and none have yet been validated as predictive biomarkers of response to immunotherapy for BTC. Overall, the various mechanisms of tumor immune evasion and expression of potential predictive biomarkers indicate that immune-directed therapies are a promising strategy in a subset of BTC patients and provide a basis for clinical trials evaluating immunotherapy in advanced BTC. 

## 4. Emerging Treatment Options with ICIs and a Potential New Firstline Therapy

### 4.1. ICI Monotherapy

The success of ICIs for malignancies such as HCC has led to an increasing number of studies evaluating their use in BTC. KEYNOTE-028 was a phase I basket trial with 20 different solid tumor cohorts, including 24 patients with advanced and relapsed BTC, who received the anti-PD-1 monoclonal antibody (mAb) pembrolizumab (10 mg/kg every 2 weeks). PD-L1 positivity, defined by ≥1% expression of tumor and tumor-associated cells, was required for trial enrollment. The ORR was 13% (3/23, all PR) with a median PFS of 1.8 months and median OS of 5.7 months. Of the three responders, two (one MSI-H, the other unknown MSI status) achieved response durations lasting more than 4 years by the time of data cutoff. The majority of patients (91.7%) discontinued treatment, primarily due to disease progression and one case related to toxicity. Common treatment-related side effects included fevers (16.7%), nausea (12.5%), and pruritis (12.7%) with no grade 4 or higher toxicities. KEYNOTE-158 was a larger phase II study that also evaluated pembrolizumab (200 mg every 3 weeks) in 104 recurrent and advanced BTC patients. Unlike KEYNOTE-028, this trial did not require PD-L1 positivity for enrollment, but retrospectively analyzed tumor biomarkers and found 61 patients (58.7%) with PD-L1-positive tumors although none were MSI-H. Positive PD-L1 expression was associated with higher ORR (6.6% vs. 2.9%), which was 5.8% (6/104, all PR) for the entire cohort. However, PD-L1 positivity did not correlate with superior survival outcomes, with no significant differences in median PFS (1.9 vs. 2.1 months) and OS (7.2 vs. 9.3 months) between PD-L1-expressing and non-expressing subgroups [82]. 

In contrast to the KEYNOTE-028 results, a phase I study evaluating nivolumab (anti-PD-1) with and without GEMCIS in a Japanese cohort with relapsed and unresectable BTC found that PD-L1 expression in at least 1% of tumor cells correlated with longer median OS (11.6 vs. 5.2 months) and PFS (2.8 vs. 1.4 months) in the nivolumab monotherapy group [82,83]. A multi-center phase II study of nivolumab (240 mg every 2 weeks for 16 weeks, then 480 mg every 4 weeks) in 46 patients with advanced refractory BTC demonstrated an ORR 22% (10/46) and DCR of 59% (27/46) by investigator assessment. Median PFS was 3.7 months and median OS was 14.2 months. None of the responders were dMMR, but positive PD-L1 status (≥1% expression) was associated with significantly longer PFS (HR 0.23, *p* < 0.001), albeit not OS. Common side effects of nivolumab included increased alkaline phosphatase, lymphopenia, and fatigue [84]. 

Durvalumab (anti-PD-L1) monotherapy has also been studied in advanced BTC. A phase I study evaluated the safety and efficacy of durvalumab with and without tremelimumab (anti-CTLA-4) in previously treated advanced BTC. In the monotherapy cohort of 42 patients, treatment with durvalumab (10 mg/kg every 2 weeks) was tolerable with no observed unexpected toxicities or treatment-related deaths. Only two patients achieved PR (4.8%) and disease control rate was 16.7% at 12 weeks. Median duration of response was 9.7 months and median OS was 8.1 months [85]. 

Bintrafusp alfa (M7824) is a novel bifunctional fusion protein that binds both TGF-β and PD-L1, thereby neutralizing their activities involved in tumor cell proliferation. Thirty patients with refractory BTC received bintrafusp alfa for a median of 8.9 weeks in a phase I open-label study. Frequent toxicities observed were rash, fever, and increased lipase. The ORR was 20% (6/30), median PFS 2.5 months, and median OS 12.7 months. Only 1 of the 6 responders was MSI-H and there was no correlation between treatment response and PD-L1 expression or TMB [86]. Based on these findings, the phase II INTR@PID BTC 047 trial evaluated bintrafusp alfa for 159 patients with platinum-refractory advanced BTC. Although an ORR of 10.1% was observed, the study did not meet the predefined threshold that would have allowed for regulatory filing for its use in the second-line setting [87]. 

### 4.2. ICI-Based Combinations

#### 4.2.1. Dual ICIs

In order to improve efficacy and overcome potential resistance with ICI monotherapy, the addition of other agents, including different ICIs, chemotherapy, and targeted agents for advanced BTC, is under investigation. Multiple preclinical studies indicate that the combination of CTLA-4 and PD-1 inhibitors is more effective than monotherapy, potentially due to synergistic effects resulting in increased numbers of TILs, decreased Tregs, and overall improved inhibition of tumor growth [88,89]. The multi-center phase II CA209-538 trial studied the combination of nivolumab plus ipilimumab (anti-CTLA-4) in advanced rare cancers including 39 patients with BTC (16 iCCA, 10 eCCA, 13 GBC). Among the BTC subgroup, the primary endpoint of disease control rate was 44%, with an ORR of 23% (9/39) and median duration of response that was not reached. None of the treatment responders had eCCA or MSI-H tumors, and they all had received prior treatment. The median PFS was 2.9 months and OS was 5.7 months [90]. A similar dual ICI regimen evaluated for relapsed and advanced BTC was durvalumab (20 mg/kg every 4 weeks) plus tremelimumab (1 mg/kg every 4 weeks) in the aforementioned durvalumab phase I study. This regimen resulted in an ORR of 10.8% (7/65, all PR), disease control rate of 32.2% at 12 weeks, and OS was 10.1 months. Grade ≥ 3 treatment-related adverse events occurred in 23% of patients [85]. 

#### 4.2.2. ICI + Chemotherapy: A New Frontline Standard of Care?

Emerging evidence indicates that cytotoxic chemotherapy works synergistically with immunotherapy by enhancing anti-tumor immunity [91]. Certain chemotherapy agents can modulate the tumor microenvironment through two major mechanisms: the induction of immunogenic tumor cell death and inhibition of mechanisms utilized by tumors for immune evasion [92]. Most chemoimmunotherapy trials in advanced BTC applied GEMCIS as the chemotherapy backbone, although oxaliplatin-based regimens were also used. EORTC-1607 (NCT03260712) and KEYNOTE-966 (NCT04003636) are ongoing phase II/III trials evaluating the addition of pembrolizumab to GEMCIS for first-line therapy in unresectable and advanced BTC. The combination of nivolumab plus GEMCIS was tested in a phase II study that enrolled 32 chemotherapy-resistant and naïve BTC patients. Two patients achieved responses (1 CR, 1 PR, ORR 33%) among the six patients in the chemotherapy-resistant cohort, whereas the ORR in the chemotherapy-naïve cohort was 61.9% (13/21, 4 CR, 9 PR). However, there were no significant differences in median PFS (3.5 vs. 6.2 months) and OS (6.7 vs. 8.6 months) between those who were previously treated versus untreated. PD-L1 expression also did not correlate with overall responses or survival outcomes, but higher TMB tumors showed a trend towards better clinical responses [93]. Final results from another similar phase II study (BilT-01) of nivolumab plus GEMCIS are pending [94]. Clinical trials evaluating nivolumab plus other chemotherapy regimens, including S-1 plus gemcitabine (NCT04172402) and nal-irinotecan plus 5-FU/leucovorin (NCT03785873) [95], are also underway. 

Toripalimab, a novel anti-PD-1 mAb, has shown safety and efficacy when combined with chemotherapy in advanced BTC. A phase II study of toripalimab plus S-1 and gemcitabine in 48 treatment-naïve BTC patients had promising survival benefits, with a median PFS of 7 months and median OS of 16 months. The ORR was 27.1% and disease control rate was 87.5% (13 PR, 29 stable disease [SD]). Biomarker analysis revealed common mutations in *TP53*, *KRAS*, *CDKN2A*, and *SMAD4*. TMB was not associated with treatment response or survival outcomes, but patients with PI3K signaling pathway activation had significantly shorter PFS. Patients generally tolerated treatment well, and frequent side effects included leukopenia, anemia, and rash [96]. Multiple phase II studies have evaluated another anti-PD-1 mAb camrelizumab plus the oxaliplatin-based regimens GEMOX and FOLFOX in untreated, advanced BTC. Response rates ranged from 16.3 to 54%, median PFS from 5.3 to 6.1 months, and median OS from 11.8 to 12.4 months [97,98]. One of the studies reported an association between PD-L1 expression and treatment response in patients who received camrelizumab plus GEMOX; ORR was 80% in patients with PD-L1 tumor proportion score (TPS) ≥ 1% versus 53.8% in PD-L1 TPS < 1%. TMB was not predictive of response and survival, but positive post-treatment circulating tumor DNA (ctDNA) correlated with shorter PFS (HR 2.83, *p* = 0.007) [97]. 

The most promising results of ICI plus chemotherapy combination regimens have come with the use of durvalumab. GEMCIS plus durvalumab, with and without tremelimumab, demonstrated significant efficacy for chemotherapy-naïve BTC in a phase II study. Forty-five of the 121 enrolled patients who received durvalumab plus GEMCIS achieved an ORR of 73.4%, DCR of 100%, and median response duration of 9.8 months. Median PFS was 11 months and OS 18.1 months in this subgroup, comparable to survival outcomes seen with the four-drug combination (median PFS 11.9 months, median OS 20.7 months). Although baseline tissue TMB did not correlate with survival benefit, PD-L1 expression after one cycle of treatment trended with improved PFS. Frequently detected mutations on analysis were in genes involved with DNA damage repair (DDR), cell cycle regulation, and genomic instability, such as *ATM*, *BRCA2*, *CDKN2A*, and *MSH2*. The most commonly observed adverse events were nausea (59.5%), pruritis (55.4%), and neutropenia (54.5%) [99]. Other chemoimmunotherapy trials with durvalumab were less successful, including the addition of paclitaxel to durvalumab plus tremelimumab, which resulted in unexpected serious anaphylactic reactions in the phase II IMMUNOBIL PRODIGE 57 trial [100]. Notably, recent data from the randomized phase III TOPAZ-1 trial demonstrated that durvalumab plus GEMCIS significantly improved survival outcomes compared to chemotherapy alone as a first-line treatment in advanced BTC. The addition of ICI to standard first-line GEMCIS resulted in superior OS (12.8 vs. 11.5 months; HR 0.80, *p* = 0.021), PFS (7.2 vs. 5.7 months; HR 0.75, *p* = 0.001), and ORR (26.7% vs. 18.7%) compared to chemotherapy alone. Patients in the experimental arm received 1500 mg of durvalumab every 3 weeks with GEMCIS for up to eight cycles, followed by durvalumab 1500 mg every 4 weeks, until disease progression or unacceptable toxicity. Treatment with the chemoimmunotherapy combination was generally well tolerated and rates of grade 3 or 4 adverse events were similar between both groups (62.7% with durvalumab vs. 64.9% with placebo) [19]. Final reports of the trial data that may identify which subgroups most benefited from immunotherapy, such as BTC subtypes, patient characteristics, and predictive biomarkers, are eagerly awaited. Based on the positive results for TOPAZ-1, GEMCIS plus durvalumab will most likely become a new standard first-line systemic therapy option for advanced BTC, signaling a pivotal change in the frontline treatment landscape more than a decade after the ABC-02 trial. 

#### 4.2.3. ICI + Anti-Angiogenic Agents

The overexpression of neo-angiogenic pathways such as VEGF is common in BTC, prompting the evaluation of angiogenesis inhibitors for these cancers [101]. Previous studies of anti-angiogenic therapies with mAb and tyrosine kinase inhibitors (TKIs) including bevacizumab, ramucirumab, sorafenib, and regorafenib, have shown mixed efficacy, both alone and with chemotherapy in untreated and relapsed advanced BTC [102,103,104,105,106]. However, the increased research focus on identifying mechanisms of resistance to ICIs has led to growing awareness of the important role that angiogenesis plays in immune suppression. Angiogenesis factors directly inhibit APCs and effector cells as well as activating inhibitory cells, including Tregs and TAMs, which, in turn, secrete factors that support angiogenesis and contribute to a highly immunosuppressive tumor microenvironment. Thus, the simultaneous blockade of immune checkpoints and angiogenesis pathways could potentially enhance anti-tumor immunity [107]. 

Pembrolizumab plus the anti-VEGFR-2 mAb ramucirumab was well tolerated but showed limited clinical activity for relapsed, advanced BTC in a phase I study [108]. The phase II LEAP-005 trial evaluated the combination of pembrolizumab plus lenvatinib, an anti-angiogenic multikinase inhibitor, in advanced solid tumors, including 31 patients with previously treated, advanced BTC. DCR was 68% (3 PR, 18 SD) and ORR was 10%, while median PFS was 6.1 months and median OS was 8.6 months. The most frequent adverse events were hypertension, dysphonia, and diarrhea. Based on these results, enrollment in the BTC cohort was expanded to 100 patients, with final analysis pending [109]. Another phase II study showed an improved median PFS of atezolizumab (anti-PD-L1) plus cobimetinib (MEK inhibitor) compared to atezolizumab alone (3.65 vs. 1.87 months); however, response rates were low in both groups (1 PR each) [110]. Other combinations of anti-angiogenic drugs and ICIs, including toripalimab plus lenvatinib (NCT04211168), are being evaluated in ongoing studies. 

The addition of chemotherapy to the combined angiogenesis/checkpoint blockade has the potential to further augment the anti-tumor immune response. A phase II study examined the combination of toripalimab and levatinib with GEMOX as first-line treatment for advanced iCCA. Among the 30 enrolled patients, the ORR was 80% (24/30), with one patient achieving CR and three patients with locally advanced tumors that were successfully downstaged and then underwent resection. The median duration of response was 9.8 months, median PFS was 10 months, and median OS, remarkably, had not been reached at a median follow-up of 16.6 months. Responses correlated with positive PD-L1 expression and DDR-related mutations. Non-hematologic side effects were jaundice (10%), rash (6.7%), and proteinuria (6.7%), with no observed grade 5 toxicities [111]. The phase II IMbrave 151 trial (NCT04677504) plans to randomize 150 patients with treatment naïve, advanced BTC to GEMCIS plus atezolizumab with or without bevacizumab (anti-VEGF). PFS per RECIST 1.1 is the primary endpoint and biomarker analysis will be performed on collected tissue, blood, and stool samples [112]. 

#### 4.2.4. Other ICI-Based Combinations

As previously mentioned, approximately 20% of BTCs express either IDH1 or FGFR2 mutations, which can be targeted by novel molecular agents. Recent early-phase studies are examining the tolerability and effectiveness of FGFR (NCT02393248) and IDH1 (NCT03684811) inhibitors combined with ICIs in advanced solid malignancies including BTC. BRCA 1/2 mutations occur in only 1–7% of BTCs, but DDR mutations have been reported in up to 63.5% of BTC tumors [113]. Tumors that harbor either of these genetic alterations seem highly vulnerable to poly ADP-ribose polymerase inibitors (PARPi), which result in genomic instability and cell death. Prior studies suggest that PARPi may promote responsiveness to ICIs by increasing neoantigens and TMB, recruiting T cells through activation of the cGAS-STING signaling pathway, and upregulating PD-L1 expression [114]. Some phase II studies are currently investigating different combinations of PARPi and ICI, such as nivolumab plus rucaparib (NCT03639935) and dostarlimab (ant-PD-1) plus niraparib (NCT04895046). Based on preclinical data supporting the potent immunoregulatory effects of epigenetic modulators, including histone deacetylase inhibitors (HDACi) and DNA methyltransferase inhibitors (DNMTi), the combination of these agents with ICIs for relapsed advanced CCA is being explored in clinical trials (NCT03257761, NCT03250273) [115]. Table 2 summarizes the major findings of clinical trials evaluating ICI-based treatments in advanced BTC.

### 4.3. Other Immunotherapy Options

#### 4.3.1. Cancer Vaccines

Cancer vaccines utilize tumor-specific antigens based on peptides and DCs to prime T cells and enhance the anti-tumor immune response. The most commonly used targets for vaccine therapy are Wilms tumor-1 (WT-1) and Mucin-1 (MUC-1), which are both overexpressed in BTC and associated with worse prognosis [19]. A phase I study of WT-1 vaccine and gemcitabine for advanced pancreatic cancer and BTC was found to have tolerable toxicity but only modest clinical efficacy, with a median OS of 9.5 months [116]. Sixty-five patients with relapsed or unresectable BTC received DC-based vaccines targeting WT-1 and MUC-1 in one retrospective study. Most patients (77%) received chemotherapy simultaneously, and the combination of chemotherapy and DC-based immunotherapy led to better response rates and survival outcomes compared to vaccines alone (HR 0.51, *p* = 0.025). The authors concluded that DC vaccines were safe, although insufficient without the addition of chemotherapy to achieve meaningful clinical responses [117]. Despite some encouraging results in early phase studies, vaccine therapies in BTC remain investigational.

#### 4.3.2. Adoptive Cell Therapy

Adoptive cell therapy is an immunotherapeutic strategy in which T cells are genetically modified to express chimeric antigen receptors (CAR) or tumor antigen-specific T cell receptors (TCR) in order to enhance their ability to recognize and kill cancer cells. The tyrosine kinase receptor EGFR is a promising therapeutic target, since it is commonly expressed in BTCs, although clinical trials of EGFR inhibitors in advanced BTC have generally been unsuccessful, as previously mentioned [44,45,46]. One phase I study enrolled 19 patients with EGFR-positive (>50%) advanced BTC, who were infused with T cells expressing EGFR-specific CAR after conditioning with nab-paclitaxel and cyclophosphamide. CAR T cell infusion was well tolerated, but cutaneous and mucosal side effects expected with anti-EGFR therapy were observed. Among the 17 evaluable patients, the disease control rate was 65%, with 1 CR achieved for 22 months and 10 SD for from 2.5 to 15 months after the first cycle. The median PFS was 4 months [118]. Another phase I study used HER2 CAR T cells in HER2-positive (>50%) advanced pancreaticobiliary malignancies, including 9 BTC. No cases of severe cytokine release storm (CRS) and treatment-related deaths were reported. DCR was 55% (1 PR, 5 SD) and median PFS was 4.8 months. [119] Adoptive cell transfer has generally not been as successful in solid tumors compared to hematologic malignancies, besides the use of sipuleucel-T in metastatic castration-resistant prostate cancer [120]. However, more studies have suggested that the addition of PD-1/PD-L1 blockade could improve the anti-tumor efficacy of CAR T cells in solid tumors, representing another potential strategy that warrants further evaluation [121]. Table 3 lists some of the many ongoing clinical trials investigating novel immunotherapeutic strategies in advanced BTC.

## 5. Future Directions

After an extended period of limited treatment options, significant progress in systemic therapy for advanced BTC has been made with the advent of targeted agents and immunotherapy. In the near future, ICIs are likely to be approved for untreated, advanced disease, in combination with chemotherapy. This should lead to more studies on the use of ICI-based treatment for earlier stages of BTC including neoadjuvant therapy for tumor downstaging, particularly for unresectable disease and in combination with locoregional therapies to increase their efficacy. Additional research will be needed to identify accurate biomarkers to predict which patient subgroups will benefit the most from immunotherapy, especially given the mixed results from prior studies of BTC using predictive biomarkers that were validated for other malignancies. As previously discussed, immune classification of the TME from tumor biopsies may help distinguish immunotherapy-responsive from resistant patient subgroups, but will need to be confirmed in prospective studies. Moreover, mechanisms of resistance to molecular targeted agents and ICIs need to be better characterized in order to develop strategies to overcome treatment resistance, such as using different combinations of agents or sequencing of drugs within the same class based on baseline or acquired mutations.

## 6. Conclusions

Over recent years, research on the underlying pathogenesis of BTC has significantly expanded and resulted in evolution of the treatment paradigm. Molecular agents targeting distinct driver mutations in specific subsets of BTC are now treatment options in the second and later-line settings. Emerging evidence has also highlighted the importance of the tumor microenvironment in modulating the anti-tumor immune response, revealing promising pathways for immunotherapeutic agents to target. Although the current first-line treatment option is GEMCIS, this is likely to change with the potential approval of durvalumab plus GEMCIS based on the TOPAZ-1 trial. Other ICI-based combinations and immunotherapy strategies, including CAR T cell therapy and cancer vaccines, represent exciting treatment approaches that are mostly investigational. Our increasing understanding of the complex TME of BTC may be the key to developing novel and effective therapies that ultimately change the bleak outcomes of these cancers.

## Figures and Tables

**Table 1 cancers-14-01748-t001:** Key findings of clinical trials for the approved systemic therapies in advanced BTC.

Trial Name	Treatment Arms	Line of Therapy	Primary Endpoint	ORR (%)	PFS (Months)	OS (Months)	HR
ABC-02 [7]	GEMCIS vs. Gemcitabine	First	OS	26.1 vs. 15.5	8 vs. 5	11.7 vs. 8.1	0.64
FUGA-BT [12]	Gemcitabine + S-1 vs. GEMCIS	First	OS	29.8 vs. 32.4	6.8 vs. 5.8	15.1 vs. 13.4	0.945
ABC-06 [13]	FOLFOX + ASC vs. ASC	Second	OS	5 vs. NR	4 vs. NR	6.2 vs. 5.3	0.69
ClarIDHy [10]	Ivosidenib vs. placebo	Second	PFS	2 vs. 0	2.1 vs. 1.4	10.8 vs. 9.7	0.69
FIGHT-202 [9]	Pemigatinib	Second	ORR	35.5	6.9	21.1	N/A
Javle et al. [14]	Infigratinib	Second	ORR	23.1	7.3	12.2	N/A

ORR, objective response rate; PFS, progression-free survival; OS, overall survival; HR, hazard ratio; N/A, not applicable.

**Table 2 cancers-14-01748-t002:** Findings of clinical trials on immunotherapy in advanced BTC.

Trial Name	Phase	Treatment Arm (s)	Line of Therapy	Primary Endpoint	ORR (%)	PFS (Months)	OS (Months)
KEYNOTE-028 [82]	I	Pembrolizumab	Second	ORR	13	1.8	5.7
KEYNOTE-158 [82]	II	Pembrolizumab	Second	ORR	5.8	2	7.4
NCT02829918 [84]	II	Nivolumab	Second	ORR	22	3.7	14.2
NCT01938612 [85]	I	Durvalumab (D) +/− Tremelimumab (T)	Second	Safety and Tolerability	4.8 in D, 10.8 in D + T	8.1 in D, 10.1 in D + T	-
NCT02699515 [86]	I	Bintrafusp alfa	Second	Safety and Tolerability	20	2.5	12.7
CA209-538 [90]	II	Nivolumab + Ipilimumab	First and Second	DCR	23	2.9	5.7
NCT03311789 [93]	II	Nivolumab + GEMCIS	First and Second	ORR	55.6	6.1	8.5
NCT03046862 [99]	II	GEMCIS + Durvalumab (3C) +/− Tremelimumab (4C)	First	ORR	73.4 in 3C, 73.3 in 4C	11 in 3C, 11.9 in 4C	18.1 in 3C, 20.7 in 4C
TOPAZ-1 [19]	III	Durvalumab + GEMCIS	First	OS	26.7 vs. 18.7	7.2 vs. 5.7	12.8 vs. 11.5 (HR 0.8)
JS001-ZS-BC001 [96]	II	Toripalimab + Gemcitabine + S-1	First	PFS, OS	27.1	7	16
NCT03486678 [97]	II	Camrelizumab + GEMOX	First	Safety, PFS	80 in PD-L1 TPS >/= 1%, 53.8 in TPS < 1%	6.1	11.8
NCT03092895 [98]	II	Camrelizumab + GEMOX or FOLFOX	First	ORR	16.3	5.3	12.4
NCT02443324 [108]	I	Pembrolizumab + Ramucirumab	Second	Safety and Tolerability	4	1.6	6.4
LEAP-005 [109]	II	Pembrolizumab + Lenvatinib	Second	Safety, ORR	10	6.1	8.6
NCT03201458 [110]	II	Atezolizumab (A) +/− Cobimetinib (C)	Second	PFS	2.8 in A, 3.3 in A + C	1.9 in A, 3.7 in A + C	-
NCT03951597 [111]	II	Toripalimab + Lenvatinib + GEMOX	First	ORR	80	10	-

ORR, objective response rate; PFS, progression-free survival; OS, overall survival; DCR, disease control rate; HR, hazard ratio.

**Table 3 cancers-14-01748-t003:** Select ongoing clinical trials on immunotherapy in advanced BTC.

Clinical Trial	Phase	Intervention	Primary Endpoint(s)	Setting	Recruitment Status
NCT04066491	II/III	GEMCIS +/- Bintrafusp alfa	OS	First	Active, not recruiting
EORTC-1607 (NCT03260712)	II	Pembrolizumab + GEMCIS	PFS	First	Active, not recruiting
KEYNOTE-966 (NCT04003636)	III	GEMCIS +/− Pembrolizumab	OS	First	Active, not recruiting
BiT-01 (NCT03101566)	II	Nivolumab + GEMCIS vs. Nivolumab + Ipilimumab	PFS	First	Active, not recruiting
NCT04172402	II	Nivolumab + Gemcitabine + TS-1	ORR	First	Active, not recruiting
NCT03785873	I/II	Nivolumab + Nal-Irinotecan	Safety and Tolerability, PFS	Second	Active, not recruiting
NCT04211168	II	Toripalimab + Lenvatinib	ORR, Rate of Adverse Events	Second	Recruiting
IMBrave 151 (NCT04677504) [112]	II	Atezolizumab + Bevacizumab + GEMCIS vs. Atezolizumab + GEMCIS	PFS	First	Active, not recruiting
FIGHT-101 (NCT02393248)	I/II	Pemigatinib + GEMCIS or Pembrolizumab or Docetaxel or Trastuzumab or INCMGA00012 in FGF/R-altered CCA	Safety and Tolerability	Second	Completed
NCT03684811	I/II	FT-2012 + Nivolumab or GEMCIS in IDH1-mutated BTC	Safety and Tolerability, ORR	Second	Active, not recruiting
NCT03639935	II	Nivolumab + Rucaparib	PFS	Maintenance after First-line Platinum-based Therapy	Recruiting
NCT04895046	II	Dostarlimab + Niraparib	PFS	Maintenance after First-line Platinum-based Therapy	Recruiting
NCT03257761	I	Durvalumab + Guadecitabine	Safety and Tolerability, ORR	Second	Active, not recruiting
NCT03250273	II	Nivolumab + Entinostat	ORR	Second	Completed
NCT03801083	II	Tumor infiltrating Lymphocytes	ORR	Second	Recruiting
NCT036337733	I/II	MUC-1 CAR T cells	DCR	First and Second	Recruiting
NCT04951141	I	Anti-GPC3 CAR T cells	Safety and Efficacy	Second	Recruiting

OS, overall survival; PFS, progression-free survival; ORR, objective response rate; DCR, disease control rate.

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
