# Peer review of "Evolving Role of Immunotherapy in Advanced Biliary Tract Cancers"

_cancers, 2022, doi:10.3390/cancers14071748_

Round 1

Reviewer 1 Report

Kang et. al., described the need of novel therapeutic agents for the treatment of BTC owing to their poor prognosis and the increasing number of incidences globally. The authors described the current treatments available for the patients especially at the late stages and the shortcoming of those approaches. With the advancement of our knowledge of genetic mutations and underlying pathways involved, new developments are being made in the field of immunotherapy including immune checkpoint inhibitors, cancer vaccines, and adoptive cell therapy in addition to molecular targeted agents. While the similar line of studies reported already, the current review, provides rational for the immunotherapy which is beneficial to the researchers working in the cancer field.

I had no specific suggestions for the authors with exception of the following:

  1. Authors are suggested to include the administrative dosage of the drugs.
  2. Authors are suggested to discuss about Derazantinib, Debio 1347, Erdafitinib as oral potent inhibitor of FGFR
  3. Authors are suggested to describe Regorafenib as an oral MKI targeting VEGFR.
  4. Authors are suggested to include ALK/ROS1 inhibitors.
  5. Authors are suggested to discuss about PIK3CA mutation as therapeutic target.

Author Response

Reviewer 1 Comments for the Author

  1. Authors are suggested to include the administrative dosage of the drugs.

- In this revision, we have included the administrative dosages of the drugs in each of the major clinical trials discussed as recommended.

  1. Authors are suggested to discuss about Derazantinib, Debio 1347, Erdafitinib as oral potent inhibitor of FGFR.

- A brief discussion regarding derazantinib, debio 1347, and erdafitinib is now incorporated in Section 2.2.2.

  1. Authors are suggested to describe Regorafenib as an oral MKI targeting VEGFR.

- Regorafenib is now mentioned under Section 4.2.3 as an anti-angiogenic TKI.

  1. Authors are suggested to include ALK/ROS1 inhibitors.

- ALK/ROS1 inhibitors are now mentioned in Section 2.2.2.

  1. Authors are suggested to discuss about PIK3CA mutation as therapeutic target.

- A short paragraph discussion drugs targeting the PI3K/AKT/mTOR pathway has been incorporated in Section 2.2.2.

Reviewer 2 Report

Dear Editor, thank you so much for inviting me to revise this manuscript about cholangiocarcinoma.

This study addresses a current topic.

The manuscript is quite well written and organized. English could be improved.

Figures and tables are comprehensive and clear.

The introduction explains in a clear and coherent manner the background of this study.

We suggest the following modifications:

  • Introduction section: although the authors correctly included important papers in this setting, we believe some studies regarding BTC immunotherapy should be cited within the introduction ( PMID: 33215952 ; PMID: 33571059 ), only for a matter of consistency. We think it might be useful to introduce the topic of this interesting study.
  • The authors should expand some sections, including a more personal perspective to reflect on. For example, they could answer the following questions – in order to facilitate the understanding of this complex topic to readers: What are the knowledge gaps and how do researchers tackle them? How do you see this area unfolding in the next 5 years? We think it would be extremely interesting for the readers.

However, we think the authors should be acknowledged for their work.

We believe this article is suitable for publication in the journal although some revisions are needed. The main strengths of this paper are that it addresses an interesting and very timely question and provides a clear answer, with some limitations.

We suggest a linguistic revision and the addition of some references for a matter of consistency. Moreover, the authors should better clarify some points.

Author Response

  1. Introduction section: although the authors correctly included important papers in this setting, we believe some studies regarding BTC immunotherapy should be cited within the introduction (PMID: 33215952 ; PMID: 33571059 ), only for a matter of consistency. We think it might be useful to introduce the topic of this interesting study.

- These references have now been incorporated within the introduction as recommended.

  1. The authors should expand some sections, including a more personal perspective to reflect on. For example, they could answer the following questions – in order to facilitate the understanding of this complex topic to readers: What are the knowledge gaps and how do researchers tackle them? How do you see this area unfolding in the next 5 years? We think it would be extremely interesting for the readers.

- Based on these recommendations, we have now included new Section 5. Future Directions. This section addresses current knowledge gaps including the need to validate predictive biomarkers and develop strategies to overcome treatment resistance.

  1. We suggest a linguistic revision and the addition of some references for a matter of consistency. Moreover, the authors should better clarify some points.

- We have edited the grammar of the paper and added references as recommended in comment 1. We have expanded upon certain points in multiple sections, including Section 3, and added an additional Future Directions section as mentioned in comment 2.

Round 2

Reviewer 2 Report

Acceptance